# A Method for Assessing the Stability of Digital Automatic Control Systems (ACS) with Discrete Elements. Hypothesis and Simulation Results

**Vladimir Kodkin \*, Alexander Baldenkov and Alexander Anikin**

Power Engineering Faculty, South Ural State University, 454080 Chelyabinsk, Russia;
baloo@mail.ru (A.B.); anikinas@susu.ru (A.A.)
\* Correspondence: kodkina2@mail.ru

**Abstract:** The article presents a new approach to the analysis of the stability of automatic systems with discrete links. In almost all modern automatic control systems (ACS), there are links that break signals in time. These are power controlled switches—transistors or thyristors operating in a pulsed mode and digital links in regulators. Time discretization significantly affects the stability of processes in the automatic control system. The theoretical analysis of such systems is rather complicated and requires a significant change in engineering approaches to analysis. With the improvement of digital controllers and a significant increase in their performance, this problem has practically been forgotten. However, its mathematical "content" has not changed since the 1980s when discreteness began to play a major role in hindering the transition to digital automatic control systems. In this paper, we propose a new approach that consists of interpreting the sampling operation by a link with the proposed frequency characteristic, which determines the suppression of input high-frequency signals. This link greatly simplifies engineering calculations and demonstrates the new capabilities of sampling systems. These possibilities include the rational distribution of digitalization resources—the number of bits and the sampling interval between the regulator channels, depending on the frequency range of the efficiency of these channels. We verify and confirm our theoretical statements through simulations and show how this approach makes it possible to formulate new principles of construction of seemingly well-known controllers—PID (Proportional Integral Differential) controllers and variable structure systems (VSS).

**Keywords:** discreteness; frequency response; automatic control system; PID controller; variable structure system with sliding processes

## 1. Introduction

Discreteness of signals "appeared" in automatic control systems (ACS) as a result of the technological improvement of regulators and power electronics. In power electronics, impulse control has made it possible to significantly increase the efficiency of converters. In control systems, digital technologies have several advantages over analog technologies, which has been thoroughly shown in the literature. Discreteness, in this case, is perceived as an inevitable problem that is solved by increasing sampling rates to the necessary level.

The sampling period for ACS is a few microseconds, and the switching period of the impulse elements is 5–10 kHz. However, the main processes for electric drives are within 10–100 Hz, and it might be logical to conclude that this unpleasant discreteness can be "forgotten" considering the attention paid to this problem in papers on digital electric drives. However, for high-precision electromechanical systems, the problems of the discreteness of information signals and power currents remain important. Indeed, the discreteness in time and in the level of the processed signals inevitably breaks the continuous ACS and makes the stability of their processes unpredictable. This is especially important

for systems that are essentially nonlinear, in which it is rather difficult to predict the reaction to all possible variations of the setting signals and disturbance factors.

Since the majority of real ACS should be referred to such systems in a refined analysis, their analysis is of great practical importance. At the same time, the generally accepted basic provisions of the analysis of discrete systems have a number of concepts that are not very acceptable for engineering analysis. Correction of these basic provisions turned out to be necessary to significantly expand the capabilities of the analysis of ACS with discrete links.

## 2. Problem Statement. State of the Issue

All methods of analysis of discrete, impulse, digital systems are in some way connected with the use of delay links and lattice functions [1-4]. These are discrete transformations of continuous signals and transfer functions—Z-transformations, D-transformations, discrete Laplace transforms and others. What these have in common (and the main thing for working with real ACS) is that all elements of the control system undergo transformations—continuous, linear, with simple and complex transfer functions. However, even then, precise transformations cannot theoretically be carried out.

This is well explained in the classic works on automatic control theory (ACT) of the 1980s. For example, in the book "Fundamentals of Automatic Control" (Moscow, "Nedra", 1972) by M.V. Meerov [1], the author states on page 332 that inverse Z-transformations: "[make] sense if the series converges...". Later in the book he speaks about calculation results, stating: "you can restrict yourself to a finite number of terms of equation (7.137)" and the frequency response of the discretization link is reduced to the characteristic of the delay link

$$W^*(j\omega) \approx \sum_{n=0}^{N} w_n e^{-j\bar{\omega}n}$$

In this case, it is stipulated that "the clock frequency is greater than the range of frequencies under consideration".

The book "*Foundations of the theory of automatic systems*" (Moscow, "Nauka", 1977) by Ya. Z. Tsypkin is very widely used in modern works on ACS [2]. The author states in paragraphs 25.3, 25.4, and 28.2: "...for sufficiently small pulse repetition periods, the impulse system can be considered as continuous, containing the same continuous part and a delay element". In other words, the structures shown in Figure 1 are equivalent.

$$W^*(j\omega) \approx e^{-j\omega\frac{T}{2}}W_H(j\omega) \tag{1}$$

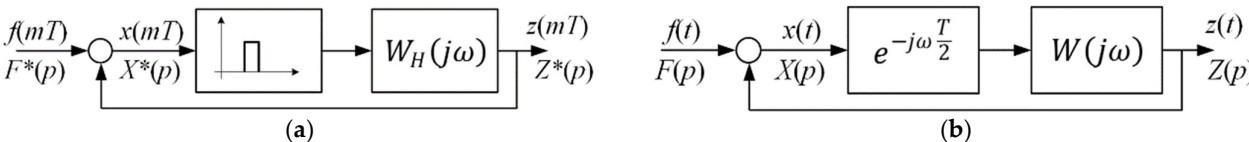

**Figure 1.** Structural diagrams of an impulse ACS (**a**) and a continuous ACS with a "delay" link (**b**).

At the same time, the author speaks about a small sampling time, although the value for "small" is not defined nor is it stated how serious of an error a non-small sampling time is. For engineers, the condition sounds something like this: "The cutoff frequency of the system must be at least 10 times less than the quantization frequency. Otherwise, nothing can be guaranteed." In addition, the delay link does not change the amplitude–frequency response, i.e., if the condition of "smallness" of the quantization interval is met, it can be forgotten altogether.

Over the past years, scientists have published a great deal of papers on these topics. The approach remains the same in nearly all of these papers; all methods are based on discrete Laplace transforms [5–10]. All of the papers contain the same assumption about the possibility of "forgetting" about the discreteness of the ACS in the stability analysis if the sampling frequency is significantly higher than the frequencies of the spectra of changes of coordinates of interest to the developers of ACS (most often the speed or angular movement). Moreover, the spectrum of signals reflecting changes in these coordinates lie within 10–100 Hz and change only slightly, while discrete elements in modern ACS have a sampling rate tens of thousands of times higher. The most serious discrete element is the power impulse element. The sampling frequency of pulse elements is 2–10 kHz, which is also significantly higher than the frequency of the spectrum of mechanical coordinates but noticeably less than the frequency of the ACS signals.

### 3. Suppression Link. Frequency Interpretation of Signal Sampling

Many years of experience with electromechanical systems, theoretical research and modeling have shown that the traditional representation of the discretization units with delays has two "drawbacks" in the analysis of the stability of the ACS [11,12]. First, the formal possibility of correcting the phase shift in systems with delay, and second, the "non-influence" of such links on the amplitude characteristics. We proposed the use of a suppression link, the main property of which is the complete suppression of input signals with a frequency higher than or equal to the sampling frequency, to overcome these weaknesses and show the influence of sampling links on the stability of the ACS.

The formula for the frequency response of the suppression link can be as follows

$$W = A(\omega)e^{j\varphi(\omega)}$$

where $A(\omega)$ and $\varphi(\omega)$ are, respectively, the amplitude and phase frequency characteristics, and have the following dependences

$$\varphi(\omega) = \begin{cases} -\dfrac{K_1 \cdot (\tau\omega)}{1 - \omega\tau}, \text{if } \omega \leq \dfrac{1}{\tau} \\ -\infty, \quad \text{if } \omega > \dfrac{1}{\tau} \end{cases} \tag{2}$$

$$A(\omega) = \begin{cases} K_2 \cdot e^{\frac{1}{\omega\tau-1}}, \text{if } \omega \leq \dfrac{1}{\tau} \\ 0, \quad \text{if } \omega > \dfrac{1}{\tau} \end{cases} \tag{3}$$

$$Lg[A(\omega)] = \begin{cases} \dfrac{K_3}{\omega\tau - 1}, \quad \text{if } \omega < \dfrac{1}{\tau} \\ -\infty, \quad \text{if } \omega > \dfrac{1}{\tau} \end{cases} \tag{4}$$

Here K1, K2, K3 are real, positive coefficients, $\omega$ is the frequency measured in radians per second.

A graph of the frequency response of the suppression link is shown in Figure 2.

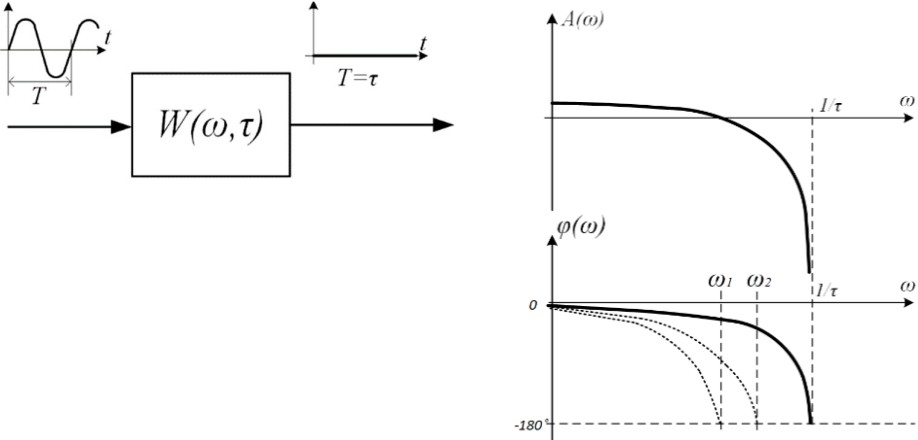

**Figure 2.** Logarithmic amplitude–frequency characteristics of the suppression link.

Formulas (2)–(4) differ from Formula (1), especially in the frequency zone close to the sampling frequency $\omega = \frac{1}{\tau}$. In this frequency zone, the signal is suppressed, and this cannot be overcome by sequential correction since no serial link will be able to overcome the suppression of the amplitude expressed by Formula (3).

The phase shift (Formula (2)), which is similar to the shift of the delay link at low frequencies, increases sharply in absolute value in the frequency zone $\omega = \frac{1}{\tau}$. It cannot be corrected in the frequency zone close to the sampling frequency.

As follows from the formulas and graphs of the frequency characteristics of the proposed suppression link, with any sequential correction at a frequency below the quantization frequency, the phase shift will reach a critical value (−180 degrees) and lead to the instability of a closed loop according to the Nyquist criteria [13] for linear ACS or V. M. Popov criteria [6,14] for nonlinear systems. The "distance" from the quantization frequency ($\omega = \frac{1}{\tau}$) at which this will occur depends on other links of the system.

Analytically, it is rather difficult to confirm the adequacy of such a representation of signal sampling operations in time, while the graphical interpretation seems to be quite clear. Modeling can provide even more clear confirmation of the effectiveness of introducing such links into the ACS to analyze the effect of signal discreteness on the ACS stability, even if they are not rigorously interpreted.

The fifteenth chapter of [15] includes consideration of several ACS models. These examples can be used to assess the capabilities of a new approach in comparison with the traditional methods of analyzing the stability of ACS. Here, "new" can refer to the sampling operation itself and to the CAP as a whole.

Let us consider the stability conditions for the second-order ACS model with proportional–integral–differential controllers (PID controllers), and variable structure systems with sliding modes (VSS with SM).

## 4. ACS with PID Controller. Frequency Response Analysis and Simulation

The block diagram of the control system of a second-order plant with a PID controller with separate controller channels (P-proportional, I-integral and D-differential channels) and different sampling intervals of these channels is shown in Figure 3. Here. $Spr_1(\tau_1), Spr_2(\tau_2), Spr_3(\tau_3)$ are suppression links with different sampling intervals.

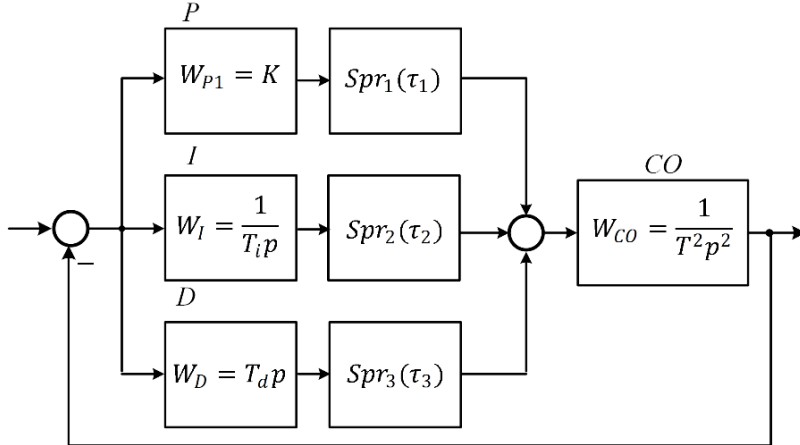

**Figure 3.** Block diagram of a second-order plant control system with a PID controller with separate controller channels (P-proportional, I-integral and D-differential channels) and different sampling intervals of these channels.

The PID controller is the most widely used controller in industrial automation. This regulator has three links, each with its own purpose and frequency range of operation:

- the **proportional** channel (P) is responsible for the system speed and the overall dynamics of the control loop; it is effective in the zone of the control loop cutoff frequency;
- the **differential** channel (D) ensures system stability and, according to the Nyquist criterion [3] and its engineering interpretations, should operate in the range from the cutoff frequency to a frequency 10 times higher;
- the **integrator** (I) provides high static accuracy of the control system, it is necessary in the zone of low frequencies, the range of its action should be less than the cutoff frequency by about 10 times, according to the same engineering stability conditions (Nyquist test).

PID controller transfer function

$$W = K + T_d p + \frac{1}{T_i p}$$

In a continuous ACS of this kind, a loop with a zero static error is easily synthesized, and the time of the transient process is 10–15 times less than the time constant of the control object.

Figure 4b shows the transients for the model shown in Figure 4a. The control object is a double integrator with an integration constant of 1 s. It should be noted that a twofold "pure" integrator is a relatively simple link for analysis, but for the stability of a closed loop system, this link, on the contrary, is quite critical. Real electromechanical links—motors, gearboxes, power elements—are of a higher order, but, as a rule, are in a range of significantly higher frequencies than the frequency range of processes in a closed loop. In addition, it should be taken into account that the frequency stability criteria work for dynamic systems with all links and interpreted frequency characteristics, including those obtained as a result of linearization. Based on the foregoing, the modeling of processes in control systems with a second-order object can be considered sufficient to illustrate the first proposed method for analyzing the stability of systems with discretization elements. In the regulator model, *Gain* is a proportional gain channel with K = 10; *Derivative* is a differentiating channel with a time constant of 2.2 s; "Transfer Fcn" is an integrating channel with a time constant of 15 s.

As follows from the diagrams in Figure 4b, the stability of the processes is close to the boundary and the time of the transient process is 1.3–1.5 s.

The links in the PID controller are connected in parallel; their resulting frequency characteristics can be determined by the "on top" rule, Figure 4a shows the logarithmic frequency characteristics (LFC) of the continuous PID controller.

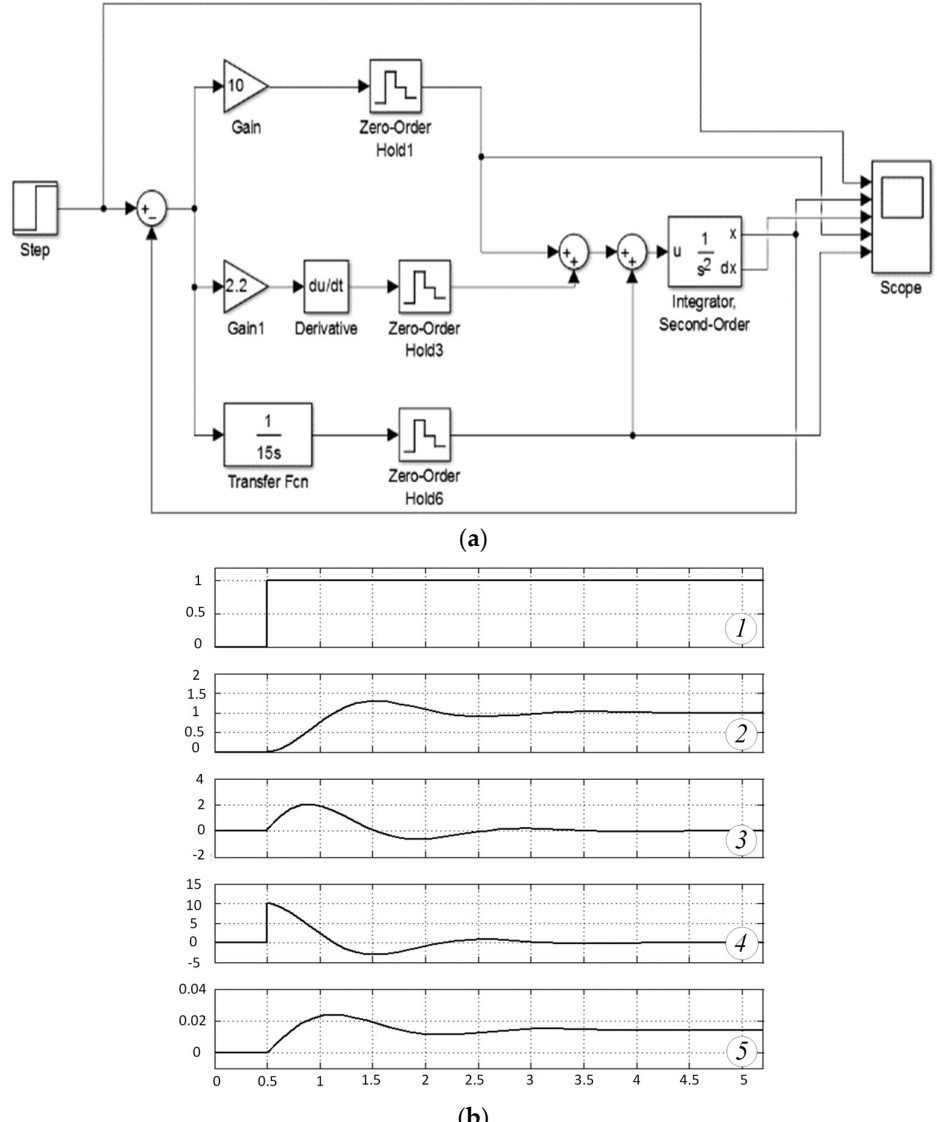

**(a)**

**(b)**

**Figure 4.** Diagram of the model with a PID controller (**a**) and diagrams of transient processes with zero delay times in the links "zero–order hold" (**b**). **1** Reference signal. **2** Controlled variable. **3** Derivative of the controlled variable. **4** Output signal of the differentiating channel of the controller after sampling. **5** Output signal of the integrator.

When adding the zero-order hold sampling links to each channel of the regulator, changes occur both in the LFC (Figure 5) and in the processes in Figure 6a,b.

As follows from the frequency characteristics in Figure 5b, the sampling frequency of the integral and proportional channels does not affect the resulting LFC of the regulator if these frequencies are higher than $\omega_1 = \frac{1}{K \cdot T_i}$ and $\omega_2 = \frac{K}{T_d}$, respectively.

Only the quantization frequency of the differential channel affects the differentiating properties of the regulator—its stabilizing "abilities". When it decreases (Figure 5c) below $\omega_2$, the differentiating properties of the regulator deteriorate significantly.

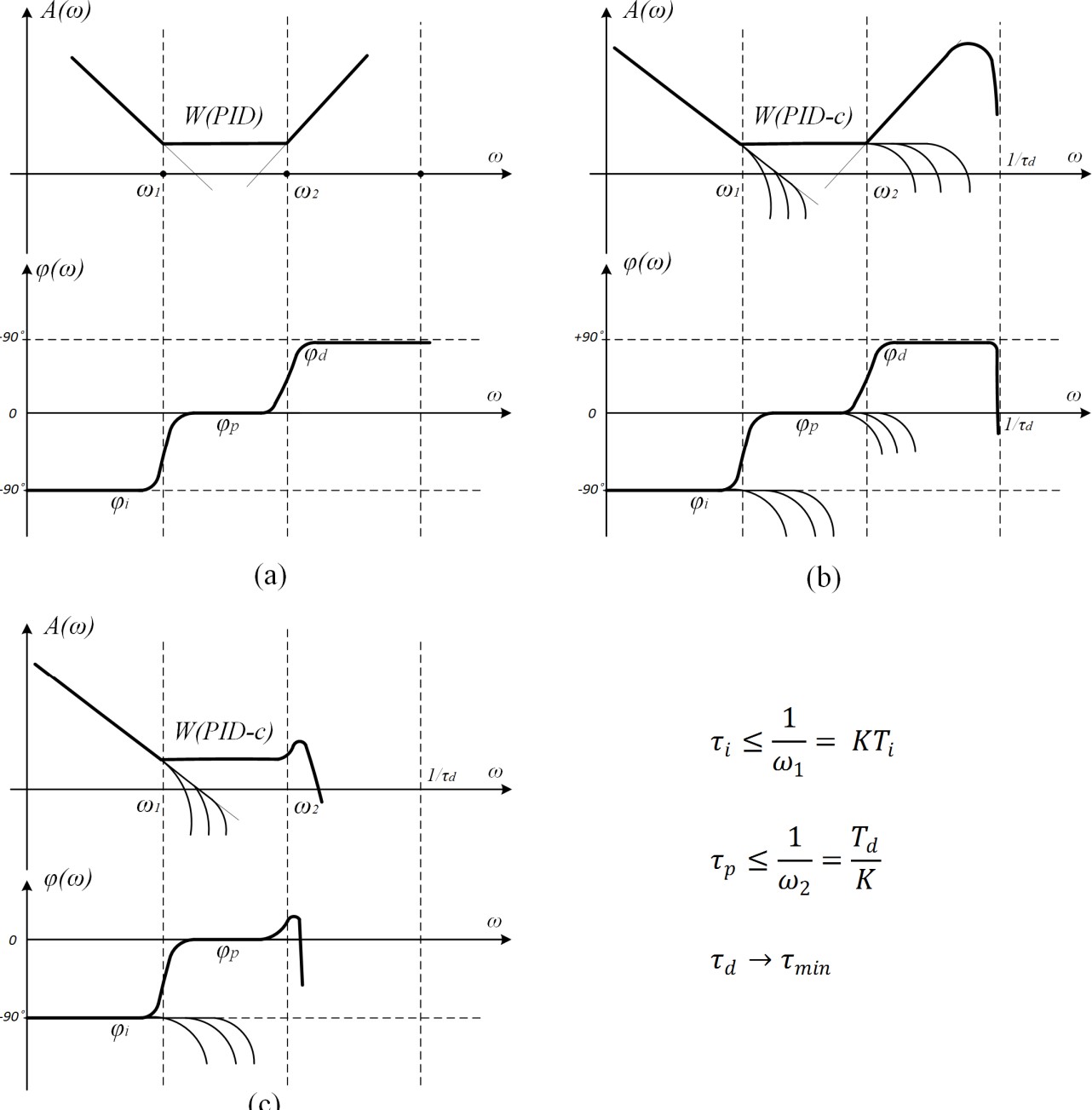

**Figure 5.** Logarithmic amplitude−frequency characteristics of the PID controller at various delays of the "zero-order hold" link.

Modeling fully confirmed these assumptions. Figure 5a,b show the processes in the model.

Usually, stability in such systems is analyzed by engineering interpretations of the Nyquist criterion and is determined by the phase shift of the frequency response of the equivalent link at the cutoff frequency of the closed loop, which determines the speed of the system [13, 16-18].

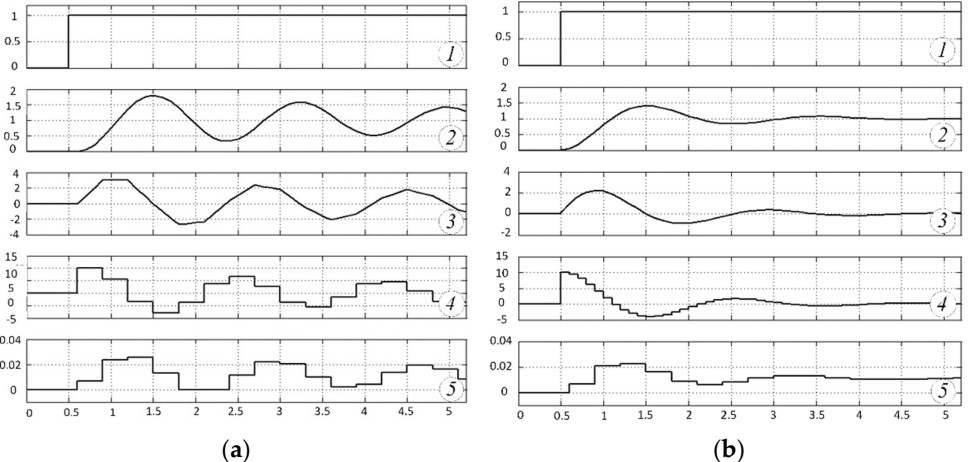

**Figure 6.** Transient processes in an automatic control system with PID controllers with a changed sampling frequency: (**a**) oscillatory processes in an ACS with a PID controller, with a total discreteness in all control channels of 0.3 s; (**b**) stable processes in the ACS with a discreteness in the D−channel of 0.1 s and discreteness of 0.3 s in the I and P control channels) **1** Reference signal. **2** Controlled variable. **3** Derivative of the controlled variable. **4** Output signal of the differentiating channel of the controller after sampling. **5** Output signal of the integrator.

After increasing the quantization time to 0.3 s in all channels, the processes became oscillatory (Figure 6a) since the PID controller became a PI controller. The LFC are shown in Figure 5c, and the processes in the model are shown in Figure 6a. In the differential channel, the discreteness is significantly reduced to 0.1 s (graph 4 in Figure 6b), and in other channels this discreteness remains the same at 0.3 s (graph 5 in Figure 6b). At sufficiently high differential channel speeds, the proportional and integral channels have practically no effect on the stability of the ACS.

The results of our stimulation confirmed the validity of the proposed interpretation of the suppression link and the possibility of using the engineering variants of the Nyquist and V.M. Popov stability criteria. The simulation also confirmed that a low sampling rate can be organized in the integral channels of PID controllers where high accuracy is required (i.e., many bits), but fast quantization in a differential channel where high accuracy is not required (i.e., the number of discharge can be reduced). This result can be called efficient multifrequency sampling. Naturally, this solution cannot be obtained with a traditional approach.

It should be noted that with different discreteness of the control channels (Figure 6b), the number of switchings in the signal at the integrator's output is significantly less than in the differential channel, therefore, less in the total signal than it would be with the same fast sampling clock in both channels. For a number of industrial control systems, reducing the number of switchings during the regulation process is a serious optimization problem since it reduces the number of switchings in equipment and increases its resource. For the sake of this increase, the equipment manufacturers recommend to go for the deterioration in the quality of regulation. When distributing the discreteness intervals according to the proposed method, the decrease in switching does not lead to a deterioration in the quality of the processes.

## 5. An example with variable structure systems (VSS)

Let us examine how the transfer function of the suppression link will "manifest" itself in systems with a variable structure with sliding processes. Figure 7 shows a block diagram of the VSS with a sliding mode (SM). Here, CO is the control object (second-order integrator) with a $W_{CO}$ transfer function, TG is a switching path generator ("slip"—$W_{TG}$), C is an amplifier ($W_C$).

In such systems, the stability condition for sliding processes is divided into the conditions for the existence of slow processes (stability condition) and the conditions for ideal sliding; that is, the existence of fast motions around slow trajectories with infinitely high frequency and infinitesimal amplitude [4,8]. Real fast movements are usually found in real ACS. Discretization can affect the fulfillment of both slip conditions. The frequency characteristics of the proposed link turn are a very convenient tool for frequency slip conditions. We previously obtained the frequency condition for the existence of sliding processes [8]. To use this condition, the concept of an equivalent circuit is introduced in Figure 8. In this circuit, the same links are connected in a slightly different way.

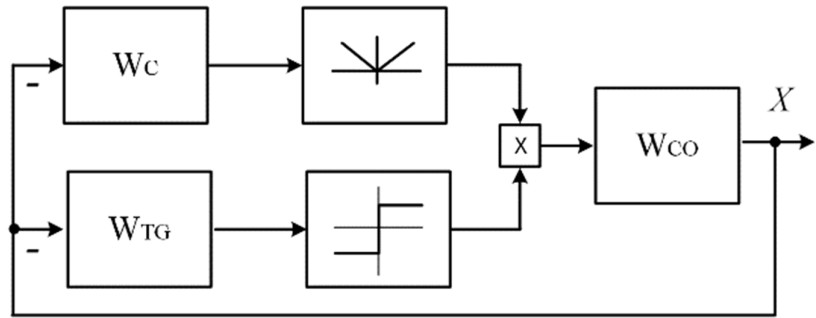

**Figure 7.** Block diagram of a system with a variable structure regulator (VSR).

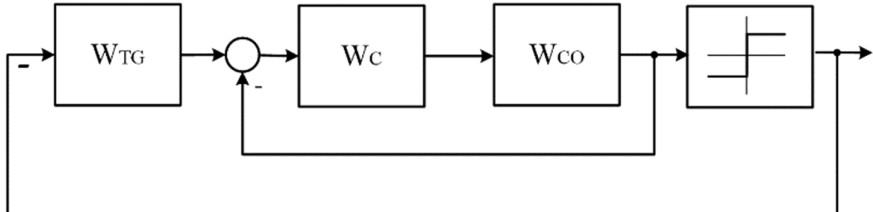

**Figure 8.** Block diagram of the substitution of VSS.

The slip condition according to the frequency characteristics of the ACS links:

The condition of ideal sliding is met when two elements—the sliding trajectory generator and the circuit formed by the controller and controlled member—are connected in series with equivalent phase characteristic of –90° minimum, and the value of –90° is reached at ω→∞.

The suggested frequency condition is met if the real part of the frequency characteristics under consideration transferred to the complex space is positive, or in a slightly different formulation

$$\text{if} \quad W_K = \frac{W_C \cdot W_{CO}}{1 + W_C \cdot W_{CO}} \tag{5}$$

then for sliding it is necessary and sufficient that the condition is fulfilled for all frequencies

$$Re[W_K \cdot W_{TG}] > 0 \tag{6}$$

$$\varphi[W_K \cdot W_{TG}] > -90^o \tag{7}$$

If this condition is not met at frequencies close to the spectrum of slow movements in the VSS, then the system is unstable; if it is not met in the frequency region 10 or more times higher, then fast sliding movements are not ideal; that is, they have a finite frequency and amplitude.

Figure 9 shows a block diagram of an SPS with sliding processes Here, the control object is a two-fold integrator, and the controller is represented by two parallel channels-

the amplifier parallel proportional and integrating links and the sliding path shaper proportional and differentiating links. The connection of the channels is carried out by the signal module extraction link and the signal sign determinant (in the simulation examples, a double integrator is used as a control object in order to make the influence of the sampling intervals of various regulator channels as clear as possible) Here, $Spr_1(\tau_1), Spr_2(\tau_2) \dots Spr_4(\tau_4)$ are suppression sections with different sampling intervals.

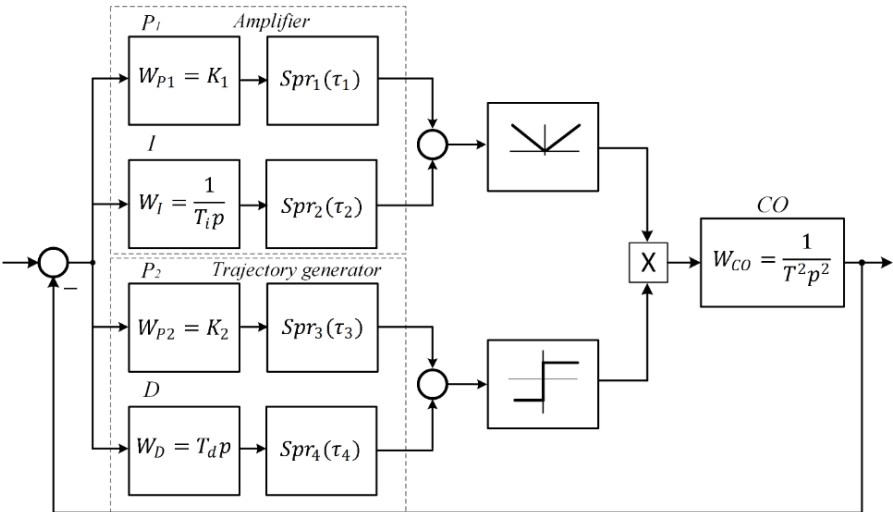

**Figure 9.** Block diagram of an VSS with sliding processes with separate sampling channels in the controller.

The diagram of the model is shown in Figure 10. Here the amplifying channel is represented by the sum of the integrating channel and the proportional link with a time constant of 10 s and a gain of 100; the sliding path is formed as a first order forcing link with a time constant of 0.15 s (Derivative link); the control object, as in the model with a PID controller, is a second-order integrator with a time constant of 1 s.

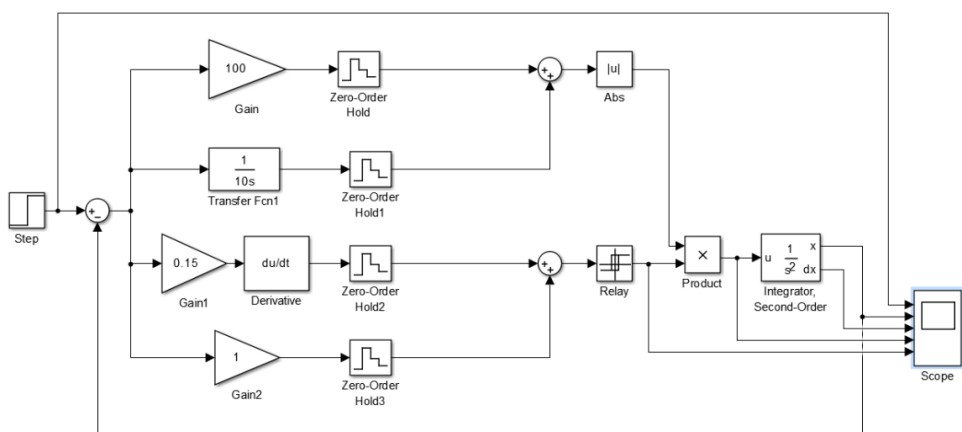

**Figure 10.** Model of a variable structure system with delay links.

Figure 11a shows the frequency response for continuous channels of the regulator. Here: $\omega_1$ is the conjugation of the characteristics of the integrator and the proportional link, $\omega_2$ is the conjugation of the characteristics of the differentiating channel and the proportional channel in the trajectory shaper, $\omega_c$ is the cutoff frequency of the slow motion contour in the VSS; Figure 11b shows the processes when all slip conditions are met.

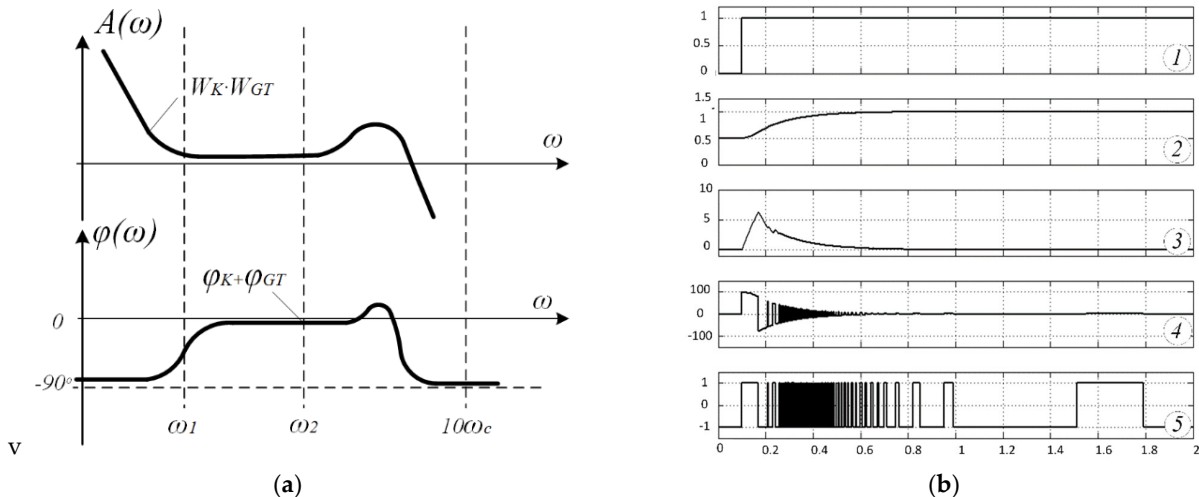

**Figure 11.** Logarithmic amplitude–frequency characteristics of VSS (**a**) and transient processes in the model with zero time delays (**b**1 Reference signal. 2 Adjustable value. 3 Derivative. 4 Signal at the output of the slip driver. 5 Output of the relay element "Relay".

With the introduction of discrete links and corresponding suppression links into the model circuit, slip conditions and processes change.

If the condition is violated in the high-frequency zone, this is a real slip. The frequency characteristics are shown in Figure 12a, and the processes are shown in Figure 12b. This violation is influenced by the discreteness of the differentiating channel of the slip driver and the proportional channel of the amplifier. This difference from a system with a PID controller is explained by the fact that in the ACS the stabilization and gain channels are not added—as in linear controllers—but multiplied (i.e., they operate in parallel). However, the discreteness of the integral channel also does not affect the processes if the simple condition $\tau_i \le \frac{T_i}{K}$ is satisfied. In this model, the parameters of the sampling units are as follows: amplification units—proportional unit—0.3 s; integrator—0.5 s; in the trajectory shaper, the sampling interval of the differential unit is 0.01 s.

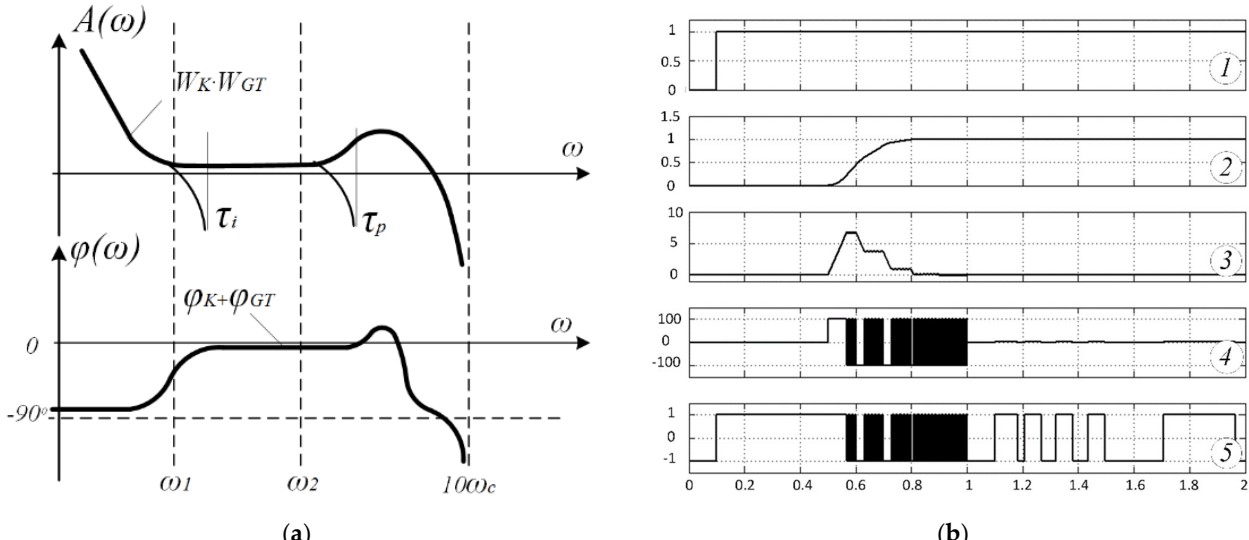

**Figure 12.** Logarithmic amplitude–frequency characteristics of VSS (**a**) and transient processes in the model with changed time delays while maintaining stability (**b**).1 Reference signal. 2 Adjustable value. 3 Derivative. 4 Derivative signal after a discrete link. 5 Relay element output.

If slip conditions, Equations (6), (7), are violated in the low-frequency zone, then there is no slip—the characteristics are shown in Figure 11a and the processes in Figure 11b. Such a state is formed in the system if the sampling interval of the differentiating channel is greater than 0.15 s. For the VSS in Figure 11, the parameters of the discretization links are as follows: gain links—proportional link, integrator and differential link—0.3 s.

Thus, the simulation has confirmed that the suppression link with the proposed frequency response fits very well into the frequency condition for the existence of slip and allows for different effective discreteness values to be set in the regulator channels.

The sample rate requirements might look like this

$$\tau_i \le KT_i; \quad \tau_p \le \frac{T_d}{K}; \quad \tau_d \le \frac{1}{10\omega_c}$$

The integral link in the amplification channel does not significantly complicate the system. As shown in Figures 9–11, the frequency response of the amplifying link is practically undistorted by the integrator sampler, and the frequency response of the trajectory shaper by the proportional channel sampler in this link is shown in Figures 11–13.

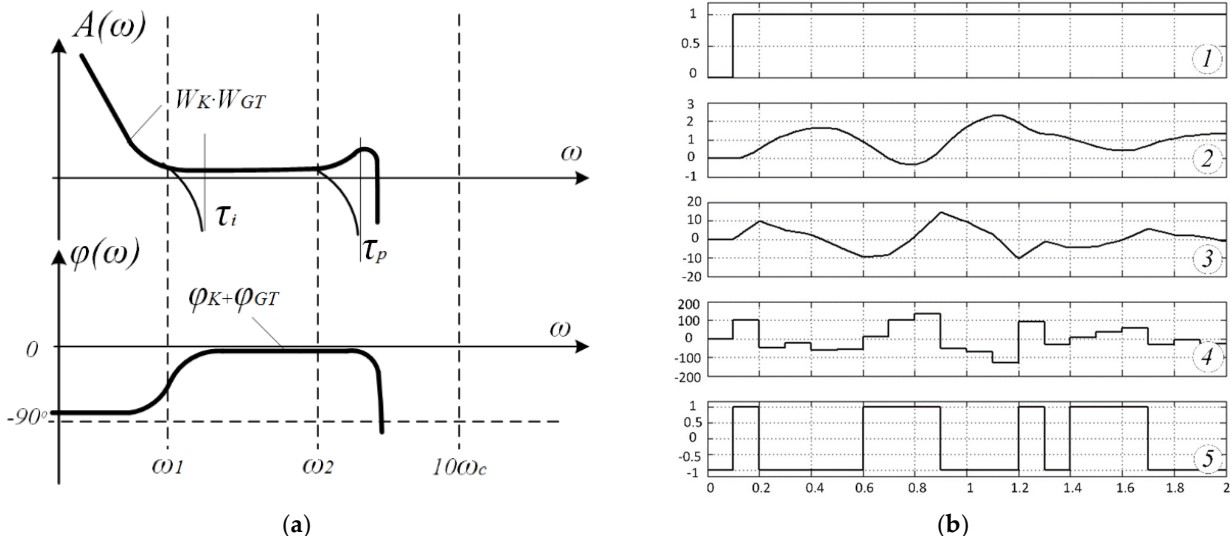

(**a**)          (**b**)

**Figure 13.** Logarithmic amplitude–frequency characteristics of VSS (**a**) and transient processes in the model with an increased time delay in the differentiation link at which the system is unstable (**b**). 1 Reference signal. 2 Adjustable value. 3 Derivative. 4 Derivative signal after a discrete link. 5 Relay element output.

That is, as in the case of the PID controller, when the links are connected in parallel, fast sampling is needed for the link operating in the high-frequency region.

For VSS with sliding processes, high-speed links are needed in the differential channel of the slip driver and in the proportional channel of the gain link. The integrator is responsible for accuracy, and can have many bits and a slow sampling rate, as in the case of the PID controller.

It should be assumed that the complication of the control object—an increase in the order of links or the presence of nonlinear structures—will not complicate the analysis of slip conditions based on the frequency characteristics of the links of the system. That is, there is reason to believe that when creating sliding processes in a complex ACS, multifrequency sampling is possible. This assumption should be verified through further independent study.

The time and nature of the processes show that the sliding processes are preserved at the speed of the channel for the formation of the slip function, which is determined in turn by the speed of the differential channel. Speed gain channels are not required but precision is. In this case, the slip condition is violated at high frequencies. This means that

fast movements are not ideal, which is consistent with the model. This confirms the validity of the previously-derived criteria for sliding along the frequency response and the effectiveness of the proposed frequency response suppression links for assessing the dynamics with nonlinear control.

The number of switchings in a system with optimized intervals can be said to be almost the same as for a simple system with PID controllers. This follows from the diagrams in Figure 6a and Figure12a . In a system with a "slow" integral channel, there will be much less of them during regulation than in SPS with one fast interval in all channels. At the same time, the quality of regulation stability, speed and accuracy will be practically the same.

When the conditions of stability (existence of sliding processes) in the VSS are rather complicated even for linear ACS, we can introduce the frequency characteristics of suppression links to obtain engineering slip conditions, even for systems with nonlinear links, and optimize the resources of regulators according to the same principles of multifrequency sampling. The same is possible for ACS with PID controllers, possibly even with greater efficiency than in continuous ACS. This is because in continuous systems with parallel connection of the links, each of them continues to work even where it is not needed. For example, in a PID controller, the integrator works in the entire frequency range. Its effect is weakened at high frequencies, but it can manifest itself negatively. The presence of a sampling link more severely restricts the operation of channels, excluding these parasitic influences.

## 6. Results Discussion

The modeling presented in this work confirms the possibility of synthesizing stable processes in automatic control systems (ACS) with links with different time discreteness, including in different channels of regulators.

In this case, the rest of the dynamic links of the system can be very different links with resonance, high-order links or nonlinearities. The only requirement for them is that they must meet the requirements applied to the dynamic links of systems, the stability of which is considered by frequency criteria, for example, the Nyquist criterion, the Mikhailov criterion or VM Popov criteria. The decisive role in influencing the stability of systems is played by the discreteness interval of the links, the frequency range of which has the most significant effect on the stability of processes in the ACS. For a system with a PID regulator, this is a differentiating channel, and for a PCA, a channel for forming a switching trajectory, this is also a differentiating link.

Modeling ACS with ATP is especially important since this is an example of a complex ACS with processes that are divided into fast and slow movements.

As shown in the article, these results are easily predicted if the discretization links are represented by dynamic links with suppression in amplitude characteristics and increasing phase lag.

The practical result following from the materials of the article is obvious, it is the ability to redistribute computing resources between individual structures. In the links responsible for the control accuracy in these examples, for integral channels the discreteness can be slow, and in the channels responsible for the stability the discreteness can be faster, while in these structures high accuracy is not required.

Another practical result follows from the analysis of the modeling processes as follows from the diagrams. The number of switchings in the output signal of the regulator shown in Figure12, in the case with different discreteness in the control channels, is much less than in a system with the same small discreteness (Figures 11 ). In a number of automatic controllers, the optimization goal is precisely the reduction in the number of switchings in the system, which inevitably worsens the regulation performance but increases the equipment time resource. The proposed method for describing the sampling links and the

method for optimizing these intervals allow us to reduce the number of switchings without deteriorating the quality of the system control, which inevitably occurs if the system has one sampling clock in all regulator channels.

In most works [1,2,10–15,16,17] devoted to the stability of systems with discretization, these links are replaced by delay elements. With this approach, the delay unit itself introduces only a phase shift, which depends on the frequency of the input signal, while the delay unit has no amplitude suppression. The amplitude characteristic of the entire system changes and becomes dependent on sampling after the transition to modified frequency characteristics, which are used in the analysis of the stability of impulse systems [2]. Moreover, it is impossible to carry out such transformations with respect to different time delays. Because of this, in such ACS, most often the most significant delay is allocated and the ACS is analyzed in the frequency domain modified with respect to this delay.

This article proposes to describe the sampling process with more "rigid" dynamic links, each of which introduces suppression of the amplitudes of the input signals at its own sampling frequency. In this case, the analysis of the stability of the entire ACS is carried out in the original frequency space, which makes it possible to take into account any number of sampling links with any sampling intervals.

The processes in the ACS models presented in the article confirm the possibility of such an approach to the analysis and to the practical synthesis of ACS with impulse elements and other sampling operations.

### 7. Conclusions

The proposed interpretation of the sampling operation in the automatic control system by a dynamic suppression link makes it possible to analyze the stability of impulse and digital systems in the base frequency space without involving the modification operation, which makes it possible to take into account the sampling links at any sampling intervals and with an unlimited number of them.

The interpretation of sampling links by more "rigid" links with the suppression of input signals proposed for assessing the stability of the ACS allows us to formulate engineering conditions for the stability of pulsed and digital systems similar to other stability criteria in frequency characteristics (Nyquist's criterion for linear control systems or Popov's criterion for nonlinear ACS). At the same time, multifrequency sampling does not complicate the analysis process but shows its practical significance, i.e., the ability to redistribute computing resources in complex digital ACS.

The simulation of ACS with multifrequency sampling, both in the simplest ACS with PID controllers and in systems with sliding processes, confirmed the correctness of its basic provisions and the fact that the discreteness of signals in time in modern electromechanical ACS is not only a minor factor, but a tool for building complex correction systems.

The proposed suppression link shows how sampling resources can be used to improve the efficiency of various regulator channels. In the channels responsible for the control accuracy (integral and amplifying links), having a high bit depth, the sampling interval may increase. In structures responsible for stability, accelerated sampling can allow a decrease in accuracy; that is, a decrease in the bit width of the regulation.

**Author Contributions:** V.K.—concept and methodology of work, project management, A.B.—development of model programs and modeling, A.A.—discussion of experimental results and analysis of materials, V.K., A.A., A.B.—editing of the article.

**Funding:** This research received no external funding.

**Conflicts of Interest:** The authors declare no conflict of interest.

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
