# Peer review of "A Method for Assessing the Stability of Digital Automatic Control Systems (ACS) with Discrete Elements. Hypothesis and Simulation Results"

_energies, doi:10.3390/en14206561_

Round 1

Reviewer 1 Report

I regret to say that is not possible to review this paper in its actual form.

Reviewer 2 Report

Please view the attachment.

Reviewer 3 Report

The manuscript presents a new approach to the analysis of the stability of automatic systems with discrete links. To demonstrate its effectiveness and applicability several simulations are draw.

Comments:

  • Manuscript is well organized and English language is fine.
  • The manuscript meets the requirements of a (state-of-the-art) literature study, authors show how the study differs from other similar algorithms/solutions and their work is put in perspective regarding advantages/disadvantages compared with others.
  • Authors should summarize the manuscript at the end of section 1.
  • Define K1 to K3 in equation 2 to 4.
  • Authors should add in label the difference between figures 5a and 5b.
  • Some paragraphs are in “Bold” style and figures labels are “highlighted in yellow”,  authors should remove them.
  • Globally the theme has potential to be published but minor changes should be done.

Author Response

 Please stt the attachment

Round 2

Reviewer 1 Report

Hi, is a very interesting paper, my first review is not fulfilled because you present a draft (I think that it was an involuntary mistake).

Now we have a complete version of your paper, I have some comments to improve it:

I do not recommend having definitions in your title (as an example automatic control systems (ACS)), neither inside of your Abstract.
Please correct: "pulsed mode and dig
ital links in regulators": * misuse of a line spacing appears in this version.
The label of Figure 3 must be corrected (appear on another page).
Some figures must be vectorized for the sake of reader convenience (at least to improve the quality of digitalization).
Figure 5 is not clear, I recommend improving with meter labeling inside of the own figure and text label. (Same for 3(b), 9(b), 10(b), 11(b))
I think that Simulink diagrams are superfluous. In any case, you can present a more general block diagram. I think that you can include it as supplementary material.
Your 5,6 and 7 equations are described in a subjective manner. I mean why 10 times? This is a typical consideration but you must explain. Same for variable structure system case.
In my point of view "slip conditions" must be explained in a better way. The actual form to explain it is confusing. 
Some paragraphs of your discussion and conclusions are repeated, please check them.
I think that sampling resources optimization is common for an engineer solving a real-world control problem. I think that you can improve your arguing talking about that fact. 

I hope that this comment can improve your paper.

Best regards...

Author Response

Comments in the app

Reviewer 2 Report

As the changes in the paper are not highlighted and the points are not addressed line by line. Please explain how the following comments from the previous round of review are addressed by directly commenting below the paragraphs. This would be helpful for the reviewer to identify the changes and offer more constructive feedbacks. Please also fix some obvious typos (e.g. There seems to be an extra break for the word “digital” in the 5th line of the abstract.)

The verification of the effectiveness for the proposed method is primarily applied to a double integrator system in simulation we a relatively large time constant on the order of seconds. In order for the method to be more generally applicable, analysis on some other types of systems (e.g. second order system with different resonance behaviors, systems with common non-linearities, etc.) and controllers (e.g. going beyond PID, what about lead/lag compensators, any suggestions on MIMO systems?) would be preferred. For the simple case on the double integrator with the PID controller, it would also be helpful to verify the system performance on a real system (e.g. implement a simple double integrator using operational amplifier circuits).

For practical consideration, the motivation of the physical significance of the system models being utilized should be highlighted more in the introduction. More discussion on applying the proposed method to practical problems would also be helpful.

There are some formatting issues with the PDF file. It seems that the paper has been reviewed previously (not by me) such that the comments and changes are highlighted. However, these information makes no sense to me as a reviewer so that displaying a final view of the paper would be preferred. The captions of the images are still not clear. For figures with multiple sub figures labeled as (a), (b), (c) such as in Figure 4, please explain the content of each sub figure in the caption. The origin of the amplitude plot axis should also be labeled. Some subscripts in math expressions should also be fixed such as in line 202.

Author Response

Comments in the app

Round 3

Reviewer 1 Report

Hi, thank you for taking my lasts recommendations.

I consider that the labeling of your figure 6 must be corrected, now is confusing.

Your figure 4 must be corrected, the scope of your diagram can be eliminated, for the sake of clarity. Maybe you can label inside of the Simulink with the number of your graph.

Best Regards...

Author Response

Comments in app

Reviewer 2 Report

Thank you for making the updates. This version addressed the previous review suggestions. Slight adjustment to the figure layouts should make them look better.

Author Response

 Comments in app

This manuscript is a resubmission of an earlier submission. The following is a list of the peer review reports and author responses from that submission.

Round 1

Reviewer 1 Report

The paper is not well written in terms of structure and presentation. The introduction is very short and differs from a common structure which includes: establishing significance, previous research and contributions; gap in knowledge and problem formulation; the present work. The conclusion is poorly formulated - it is not clear what is the paper’s contribution to knowledge. English in many paragraphs is not good – it is so difficult to understand what the authors would like to explain. I am not sure that the paper can be accepted in the present form.

The other comments are:

The abstract should be one paragraph.

Line 21-22: Two “which” in the same sentence; please revise.

Line 35: “impulse control”? – “pulse width modulation control” or “switching operation”

Line 42: “unpleasant”!?

Lines 59-60; Lines 68-69: No need to mention book title, publisher and author in the text. Please use a reference number. Please also consider the reference replacement – the sources are not available for readers (it seems the books are available in Russian only; no translation in English).

Something wrong with the figure 1 presentation.

Line 79: “… author speaks about …”?

Lines 65; 109; 156; 190; 192; 194; 325: The equations are not numbered.

Line: 123-141: the paragraph first lines have no indentations.

Figure 3(b); Figure 5; Figure 9(b); Figure 10(b); Figure 11: There are no axes titles and units.

Figure 6, Figure 7, Figure 8: There are no figure descriptions.

Lines 88-91; 103-106; 205-208; 225-228; 241-244; 350-354; 362-366: Too long sentences – please revise and brake on a number of shorter sentences.

Reviewer 2 Report

The paper presents a new approach to the analysis of the stability of automatic systems with discrete links. Unfortunately, the paper is not well written and lacks any scientific contribution and there is no literature review, nor methodoly nor propoer scietific results. There are many sentences in the paper which are not valid for example the following sentences a seem to be not accurate:

 The sampling period for ACS is a few microseconds, and the switching period of the impulse elements is 5–10 kHz. But the main processes for electric drives are within 10–41 100 Hz.

Reviewer 3 Report

The paper is well set, and the problem highlighted executed properly. However, attention should be given to the following highlighted points before resubmitting.

  1. Why is the summary divided into so many paragraphs? Please write it in one paragraph.
  2. Discussion sectionshould be included in the manuscript. The authors should present their findings and their main implications in the "Discussion" section, also highlighting current limitations of their study, and briefly mention some precise directions that they intend to follow in their future research work.
  3. The font size of the formula is inconsistent, please correct it.
  4. For material presented to flow I suggest table of symbols.
  5. It seems that all figures are missing captions, please add them.
  6. I suggest adding a description of the main contributions of this paper in I
  7. Where is the literature review? The number of literature is also too small. Please add and improve them.
  8. There are many formatting errors. For example, why is the font in line 77 marked yellow, and why is the font in line 108 bolded?There are formatting errors in lines 110-111, 116, 123-124, and many others. Please check and revise the whole article.
  9. In the Conclusionsection, the authors should avoid simply summarizing the aspects that they have already stated in the body of the manuscript. Instead, they should interpret their findings at a higher level of abstraction than in the previous sections of the manuscript.